🔓 | **Open Peer Review** | Genomics and Proteomics | Observation

# The effect of site-specific recombinases XerCD on the removal of over-replicated chromosomal DNA through outer membrane vesicles in bacteria

Johannes Mansky,[1] Hui Wang,[1] Irene Wagner-Döbler,[1] Jürgen Tomasch[2]

**ABSTRACT** Outer membrane vesicles (OMVs) are universally produced by Gram-negative bacteria and play important roles in symbiotic and pathogenic interactions. The DNA from the lumen of OMVs from the Alphaproteobacterium *Dinoroseobacter shibae* was previously shown to be enriched for the region around the terminus of replication *ter* and specifically for the recognition sequence *dif* of the two site-specific recombinases XerCD. These enzymes are highly conserved in bacteria and play an important role in the last phase of cell division. Here, we show that a similar enrichment of *ter* and *dif* is found in the DNA inside OMVs from *Prochlorococcus marinus*, *Pseudomonas aeruginosa*, *Vibrio cholerae,* and *Escherichia coli*. The deletion of *xerC* or *xerD* in *E. coli* reduced the enrichment peak directly at the *dif* sequence, while the enriched DNA region around *ter* became broader, demonstrating that either enzyme influences the DNA content inside the lumen of OMVs. We propose that the intra-vesicle DNA originated from over-replication repair and the XerCD enzymes might play a role in this process, providing them with a new function in addition to resolving chromosome dimers.

**IMPORTANCE** Imprecise termination of replication can lead to over-replicated parts of bacterial chromosomes that have to be excised and removed from the dividing cell. The underlying mechanism is poorly understood. Our data show that outer membrane vesicles (OMVs) from diverse Gram-negative bacteria are enriched for DNA around the terminus of replication *ter* and the site-specific XerCD recombinases influence this enrichment. Clearing the divisome from over-replicated parts of the bacterial chromosome might be a so far unrecognized and conserved function of OMVs.

**KEYWORDS** DNA replication, DNA repair, outer membrane vesicles

Membrane vesicles are excreted by cells from all domains of life, and their cargo and the physiological roles discovered until now are as diverse as life itself (1, 2). Outer membrane vesicles (OMVs) of Gram-negative bacteria have often been found to contain DNA, for example, in *Acinetobacter baylyi* (3), *Ahrensia kielensis* (4), *Francisella novicida* (5), *Haemophilus influenza* (6), *Kingella kingae* (7), *Moraxella catarrhalis* (8), *Prochlorococcus* sp. (9), *Pseudoalteromonas marina* (4), *Porphyromonas gingivalis* (10), and *Shewanella vesiculosa* (11–13). *Prochlorococcus marinus,* one of the most abundant species in the ocean, continuously excreted two to five OMVs per cell per generation. Here, an enrichment of the region around the terminus of replication (*ter*) in vesicle DNA was noted for the first time, suggesting a link with the cell cycle (9). In *Vibrio cholerae*, both chromosomes were found in the DNA from the vesicle lumen (14). In *Pseudomonas aeruginosa*, OMVs from planktonic cultures contained plasmids (15) and chromosomal DNA (16). Plasmids were also incorporated into OMVs by *Acinetobacter baylyi* and *Acinetobacter baumannii* and could be transferred into *Escherichia coli* (3, 17).

Address correspondence to Jürgen Tomasch, tomasch@alga.cz.

The authors declare no conflict of interest.

See the funding table on p. 8.

Gene transfer represents an important function of OMVs, e.g., by mediating the transfer of antibiotic resistance genes (18–20). In the cited studies, vesicles were always treated with DNase to remove extra-vesicle DNA. In *E. coli,* it was shown already in 1978 that vesicles are continuously produced during growth (21) and contain proteins from the outer membrane and the periplasmic space (22). While there are numerous studies on the protein content of *E. coli* OMVs, studies on the DNA cargo are rare and focused on the transfer of plasmids (23–25).

OMVs are generated by blebbing from the outer membrane and enclose molecules from the periplasmic space, which is free of DNA; it is, therefore, an unsolved question how the DNA inside the vesicle lumen was transferred from the cytosol to the periplasmic space or into the vesicle lumen, respectively (26–29). So-called outer-inner-membrane vesicles have been found in addition to "normal" OMVs in *Shewanella oneidensis* and were suggested as a possible solution (12). Another alternative is the so-called "explosive cell lysis" observed in biofilms of *P. aeruginosa* (30). In those biofilms, no blebbing of outer membranes was observed. By contrast, a subpopulation of cells in the biofilm lysed upon stress, and the shattered membrane fragments spontaneously formed small vesicles incorporating cytoplasmic DNA; this type of vesicle formation required the endolysin *lys* (30).

We had previously shown that the Alphaproteobacterium *Dinoroseobacter shibae* secretes DNA-containing OMVs constitutively during growth (31). Time-lapse microscopy captured instances of multiple OMV production at the septum of dividing cells (31). We compared the proteome of vesicles to that of cells (membrane and soluble fraction) and found that the vesicle proteome was clearly dominated by the outer membrane and periplasmic proteins. The most abundant vesicle membrane proteins were predicted to be required for direct interaction with peptidoglycan during cell division (LysM, Tol-Pal, SpoI, and lytic murein transglycosylase) (31). A metabolome analysis of OMV membranes found that they were 15-fold enriched for the saturated fatty acid 16:00, making them more rigid compared to the cytoplasmic membrane (31). DNA from the vesicle lumen was up to 22-fold enriched for the region around the terminus of replication (*ter*). The peak of coverage was located at *dif*, a conserved 28-bp palindromic sequence required for binding of the site-specific tyrosine recombinases XerC/XerD. These recombinases are activated by FtsK in the divisome complex right before septum formation, and they are known to resolve chromosome dimers (32–37). We hypothesized that constitutive OMV secretion in *D. shibae* is coupled to cell division and that these vesicles remove over-replicated chromosomal DNA at the end of the cell cycle, which would otherwise halt cell division and thus be lethal to the cell. The enrichment of *dif* points toward a role of XerCD in this process.

To test our hypothesis further, we reanalyzed the DNA content of vesicles previously isolated from the model organisms *Prochlorococcus marinus* (9), *Pseudomonas aeruginosa* (30), and *Vibrio cholerae* (14). Furthermore, we chose *Escherichia coli* as an additional model for OMV production because it is the archetypical, best-understood organism regarding replication and cell division (38, 39) and a library of well-characterized gene knockouts is available, including *xerC* and *xerD* (40). We studied two questions: (i) Is the enrichment of the *dif* site specific for *D. shibae*, an Alphaproteobacterium from the Roseobacter group, or does it occur in other bacteria as well? (ii) Are the XerCD enzymes influencing the enrichment of *ter* and *dif* in the DNA inside OMVs? When these enzymes are resolving chromosome dimers, no fragments containing *dif* are produced. Therefore, we investigated the DNA composition in the lumen of OMVs produced by deletion mutants of *xerC* and *xerD* in *E. coli*.

## Bacterial strains analyzed

An overview of all analyzed strains can be found in Table 1. Data for the *P. marinus*, *P. aeruginosa,* and *V. cholerae* vesicle DNA were downloaded from the NCBI sequence read archive. The *dif* sites were obtained from the literature (9, 32, 33). *D. shibae* DSM16493 was obtained from the DSMZ, Braunschweig, Germany. Strains *E. coli* K-12 BW251113

**TABLE 1** Strain information, mapped reads to the whole genome (total) and terminus (*ter*), summary statistics for mappings to 200 random locations, and enrichment of *ter*-located reads compared to the median along the chromosome[a,b]

| Strain | Replicon | *ter* start | *ter* end | Mapped reads | | Mean | Median | Std. dev. | Enrichment | Data accession | Reference |
|---|---|---|---|---|---|---|---|---|---|---|---|
| | | | | Total | ter | | | | | | |
| Prochlorococcus sp. Med4 | NC_005072.1 | 826,000 | 832,000 | 1,368,548 | 92,377 | 3,050 | 0 | 13,221 | 92,377 | SRR1013844 | (9) |
| Prochlorococcus sp. Med4 | NC_005072.1 | 826,000 | 832,000 | 2,729,515 | 186,179 | 16,381 | 0 | 60,827 | 186,179 | SRR1013875 | |
| Pseudomonas aeruginosa | NC_002516.2 | 2,440,067 | 2,446,067 | 55,750,742 | 54,594 | 35,740 | 49,237 | 26,894 | 1 | SRR1654902 | (30) |
| Vibrio cholerae Chr 1 | NC_009457.1 | 1,126,240 | 1,132,240 | 792,045 | 11,395 | 1,170 | 1,234 | 1,133 | 9 | SRR10387914 | (14) |
| Vibrio cholerae Chr2 | NC_009456.1 | 564,632 | 570,632 | 4,869,589 | 5,046 | 8,735 | 540 | 34,823 | 9 | SRR10387914 | |
| Dinoroseobacter shibae | NC_009952.1 | 1,613,200 | 1,620,200 | 6,129,709 | 234,919 | 11,238 | 3,122 | 32,204 | 75 | SAMEA114558114 | This study |
| Dinoroseobacter shibae | NC_009952.1 | 1,613,200 | 1,620,200 | 5,333,124 | 191,321 | 7,894 | 3,640 | 18,346 | 53 | SAMEA114558116 | |
| Escherichia coli BW25113 | NZ_CP009273.1 | 1,582,052 | 1,588,052 | 1,725,436 | 28,376 | 1,626 | 2,051 | 1,103 | 14 | SAMEA113533507 | |
| Escherichia coli BW25113 | NZ_CP009273.1 | 1,582,052 | 1,588,052 | 1,237,778 | 100,698 | 1,015 | 1,258 | 813 | 80 | SAMEA113533508 | |
| Escherichia coli BW25113 | NZ_CP009273.1 | 1,582,052 | 1,588,052 | 1,229,730 | 79,686 | 1,388 | 1,320 | 2,038 | 60 | SAMEA113533509 | |
| Escherichia coli BW25113 ΔxerC | NZ_CP009273.1 | 1,582,052 | 1,588,052 | 2,589,912 | 94,403 | 4,828 | 2,644 | 12,692 | 36 | SAMEA113533510 | |
| Escherichia coli BW25113 ΔxerC | NZ_CP009273.1 | 1,582,052 | 1,588,052 | 2,593,008 | 98,750 | 2,937 | 2,609 | 4,860 | 38 | SAMEA113533511 | |
| Escherichia coli BW25113 ΔxerD | NZ_CP009273.1 | 1,582,052 | 1,588,052 | 2,191,086 | 69,473 | 2,202 | 2,057 | 3,079 | 34 | SAMEA113533512 | |
| Escherichia coli BW25113 ΔxerD | NZ_CP009273.1 | 1,582,052 | 1,588,052 | 3,204,824 | 178,388 | 6,230 | 2,631 | 18,814 | 68 | SAMEA113533513 | |
| Escherichia coli BW25113 ΔxerD | NZ_CP009273.1 | 1,582,052 | 1,588,052 | 2,544,678 | 182,057 | 5,042 | 1,598 | 16,863 | 114 | SAMEA113533514 | |

[a]Mean, median and standard deviation were calculated from counting the reads mapped to random 6 kb regions excluding ter on the respective chromosome.
[b]Accession numbers are for the NCBI sequence read archive (SRR) or the EMBL ENA archive (SAM).

(WT), *E. coli* JW3784 (Δ*xerC*), and *E. coli* JW2862 (Δ*xerD*) were obtained from the Keio Collection (40).

## Purification of vesicles and isolation of DNA

Purification of *D. shibae* vesicles and sequencing of their DNA content were reproduced in the current study according to the previously published protocol (31). *E. coli* strains were grown on Lysogeny broth (LB) plates or liquid LB medium at 37°C, with liquid cultures shaken at 180 rpm. Cell count was determined by flow cytometry using a MacsQuant Analyzer 10, and vesicle count was determined using the NanoSight NS300 (Malvern Panalytical). Vesicles were purified from 1 L of culture per replicate. Bacterial cells were separated by centrifugation at 10,900 *g* for 15 min; the supernatant was filtered using 0.45 and 0.22 µm bottle top filters (Millipore). The filtrate was concentrated using a tangential flow filtration system (Vivaflow 200, Sartorius). The concentrate was ultracentrifuged at 100,000 *g* for 2 h at 4°C; the resulting pellets were stored at −20°C until DNA isolation.

To isolate DNA, the vesicle pellet was suspended in 176 µL sterile Phosphate-buffered saline (PH = 7.2). To remove extra-cellular DNA, 20 µL 10× DNase buffer and 4 µL DNase I (NEB Inc.) were added and incubated at 37°C for 30 min; then, the enzyme was inactivated by incubation at 75°C for 10 min. The mixture was cooled on ice for 5 min; the OMVs were lysed by the addition of 2 µL 100× GES lysis buffer [5 M guanidinium thiocyanate, 100 mM EDTA, and 0.5% (wt/vol) sarcosyl] and incubation at 37°C for 30 min. To remove RNA, 2 µL RNAse (Thermo Scientific) was added, and the sample was again incubated at 37°C for 30 min. Two hundred-microliter phenol–chloroform–isoamyl alcohol was added, vortexed, and centrifuged at 12,000 *g* for 5 min at 4°C. The upper aqueous phase was withdrawn; 200 µL TE buffer was added to the organic phase, thoroughly mixed, and centrifuged at 12,000 *g* for 5 min at 4°C. The resulting aqueous phase was removed and combined with the previously collected phase. DNA was precipitated by the addition of 40 µL of 3 M sodium acetate, 1 µL of glycogen, and 1.6 mL ice-cold ethanol. After the overnight incubation, the sample was centrifuged at 12,000 *g* for 5 min at 4°C. The supernatant was removed, and the pellet was washed three times with 70% ethanol. Afterward, the remaining ethanol was removed; the pellet was air-dried and dissolved in 20 µL TE buffer. Isolated DNA was stored at −80°C until further analysis.

## DNA sequencing and analysis

Libraries for sequencing were prepared with the NEBNext Ultra II FS DNA kit according to the manufacturer's protocol. Fifty-base pair paired-end sequencing was performed on the NovaSeq 6000 to a depth of 2 million reads per sample. Quality trimming of raw reads was conducted with sickle v.1.33. Processing and analysis of sequencing data were performed as described (41). Trimmed reads were mapped to the genome using Bowtie2 (42). To test for an enrichment of the ter region in vesicle DNA, the counts of read mapping within and outside the ter region, defined as 8 kb surrounding the *dif* site, were calculated using samtools within a custom shell script. The chromosome outside the ter region was split into 10 equal parts, and 20 samples of 8 kb within each segment were counted. Mean, median, and standard deviation were calculated from these 200 samples. The coverage per nucleotide was calculated using BEDtools (43) and summarized for sliding windows of 8 kb along the chromosome using the zoo package in R (44). For the determination of significant differences in coverage between the *E. coli* wild-type and mutant strains, edgeR (45) was employed on trimmed mean of M-values (TMM)-normalized read coverage for a window from 1.3 to 1.9 Mb. For this range, the values are normally distributed; for the full range of the chromosome, they are not. Scripts can be found on github (https://github.com/Juergent79/membrane_vesicles).

## DNA content of bacterial OMVs from published data sets

Bacterial vesicle DNA from the three published data sets showed an enrichment of the ter region although to a different extent (Table 1). The first sequencing of DNA from

membrane vesicles was reported for *P. marinus* (9) and reanalyzed here. The vesicles were produced constitutively during the exponential growth of the bacterium. In DNA from vesicles harvested from growing cells, a broader 100-kb region around *ter* was enriched with several distinct peaks, the highest located directly at the *dif* site (Fig. 1A). For *P. aeruginosa,* the sequenced DNA reportedly originated mainly from OMVs formed in biofilms during explosive cell lysis (30). In this process, the whole cellular DNA content is released and can be attached to the surface or included in the lumen of newly formed vesicles. It is, therefore, expected that over-replicated DNA from the last stage of cell division might not be particularly enriched. Indeed, the whole chromosome was represented with the highest coverage around *ori* indicating the release of DNA from replicating cells. However, a small peak could also be identified in direct proximity to the *dif* site (Fig. 1B). Vesicles of the third model *Vibrio cholerae* were isolated at the early exponential phase when cell lysis was reportedly minimal (14). The *V. cholerae* genome consists of two chromosomes. Both of them were completely covered by vesicle DNA, with their *dif* sites at *ter* ninefold enriched compared to the remainder of the chromosome and found among the highest of several distinct peaks (Fig. 1C). One phage region on chromosome 2 showed a coverage around 150,000-fold higher than the rest of the genome. This shows that part of the DNA originated from the active k139 phage encoded in this region. In summary, all three data sets indicate the enrichment of *dif* site DNA in OMVs of the respective bacteria.

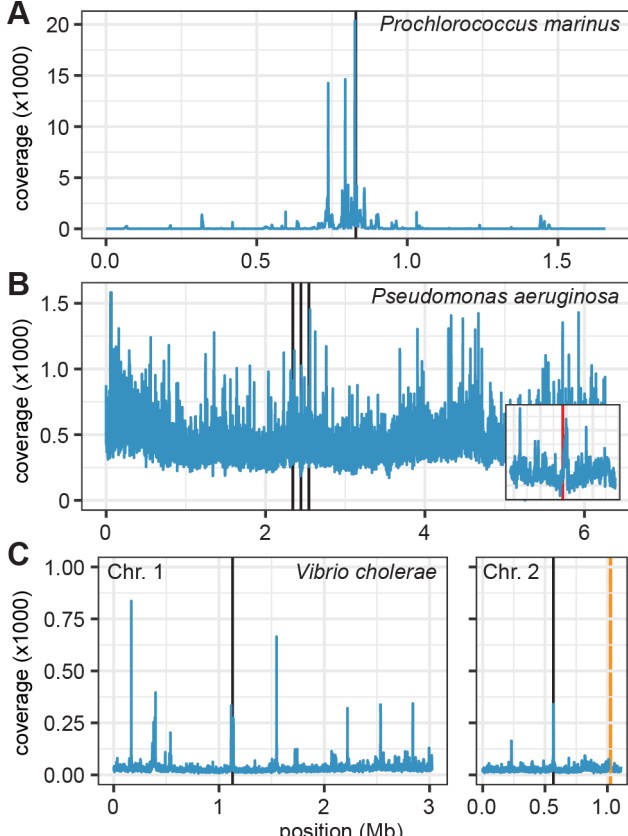

**FIG 1** DNA content of OMVs from various bacteria. Coverage of mapped reads on the chromosomes averaged for sliding windows of 0.5 kb. The *dif* site is marked in black. (A) *Prochlorococcus marinus*. (B) *Pseudomonas aeruginosa* biofilms. The inset shows the region between the outer black lines in the main figure; the *dif* site is marked in red. (C) *Vibrio cholerae* chromosomes 1 and 2. The highly enriched phage region marked in yellow on chromosome 2 has been removed from the visualization.

## Influence of *xerC* and *xerD* knockouts on the DNA content of *E. coli* OMVs

The *E. coli* Δ*xerC* and Δ*xerD* mutants grew at the same rate as the wild type (Fig. 2A); thus, they did not have an obvious fitness defect, in accordance with the published strain descriptions (40). However, the dynamics of OMV production was different in the mutants. While the OMV concentration in the supernatant remained stable around 2 $\times$ $10^8$ vesicles/mL for the wild type, it increased from a similar initial value to 5 $\times$ $10^9$ vesicles/mL for the mutants during the 12 h of cultivation (Fig. 2B). The ratio of vesicles per cell was similar for the wild type and mutants during the first 2 h of growth. Then, at 4 h, it dropped to 0.6–0.2 for the wild type while it remained between 3 and 10 for both mutants (Fig. 2C). If our hypothesis is true and the DNA in OMVs represents excised over-replicated fragments, then more such waste was produced in the mutants.

For all three strains, we found the *ter* region over-represented in the DNA isolated from the OMV's lumen (Table 1). In the wild type, a 100 kb region around *ter* and particularly the *dif* sequence almost in the center was enriched more than 120-fold compared to the rest of the chromosome (Fig. 3A and B). This is comparable to the 200-kb region surrounding the homolog site in OMVs of *D. shibae,* which was also found up to 120-fold enriched (31). The enrichment of the *ter* region in DNA of OMVs from either mutant clearly differed from that of the wild type (Fig. 3B). The peak range increased asymmetrically to approximately 350 kb around *dif* with this broader region being up to fourfold higher present in the mutant OMVs, suggesting increased and lengthened over-replication in these strains (Fig. 3B and C). A single-nucleotide-view on the most strongly enriched region revealed three peaks with the central maximum at the 28-bp *dif* site for the wild type (Fig. 3B). This maximum was 2.5-fold reduced in both mutants, while the surrounding two peaks were still visible, particularly in Δ*xerC*. Since the XerCD–FtsK–complex cannot be formed when either *xerC* or *xerD* are knocked out, these data reflect the activity of the remaining recombinase homolog.

## DNA composition of OMVs

For *P. marinus*, *D. shibae*, *V. cholerae,* and the newly analyzed *E. coli* strains, the vesicles were treated with DNase prior to analyzing the DNA inside the vesicle lumen. However, the effectiveness of DNAse treatment plays a large role in the enrichment of protected DNA, and a complete removal of extra-vesicular DNA cannot be guaranteed (46). For *D. shibae,* we previously sequenced DNA from both DNase-treated and -untreated vesicle enrichments and could show that the digestion of unprotected DNA results in a reduction of read mapping outside the *ter* region (31). In the case of *V. cholerae*, the sampling time point was chosen to minimize DNA originating from lysed cells, and two consecutive digestion steps were performed (14). In addition to the enrichment of the *ter* region, some other short specific regions and in particular phage DNA were found to be over-represented. DNA within a phage is shielded from DNase activity (41). The

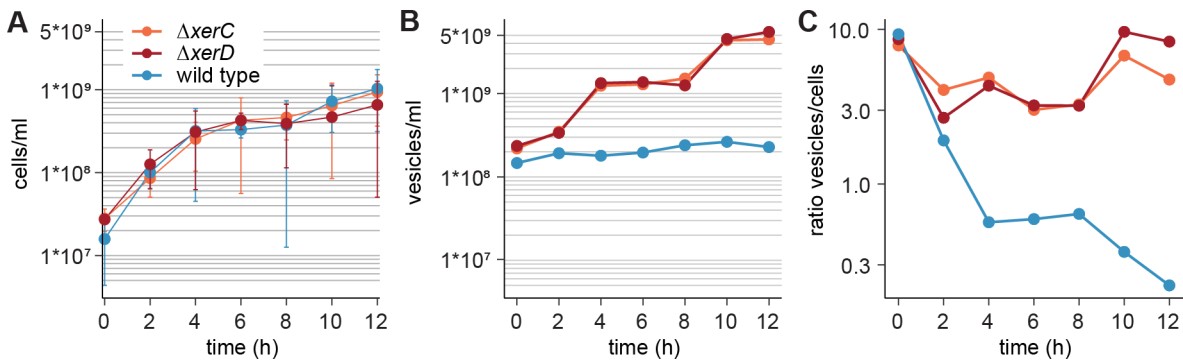

**FIG 2** Growth and outer membrane vesicle production of *E. coli*. (A) Growth of *E. coli* wild-type and mutant strains. (B) Vesicles in the supernatant of *E. coli* strains during growth. (C) Ratio of vesicles per cell during growth.

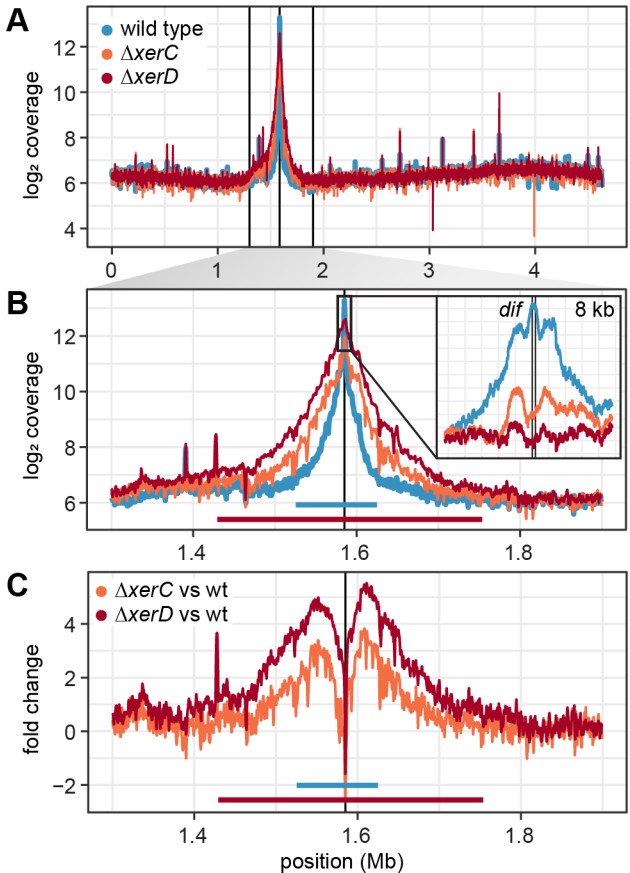

**FIG 3** DNA content of *E. coli* outer membrane vesicles. (A) Coverage of mapped reads on the chromosome of *E. coli* averaged for sliding windows of 0.5 kb. (B) Zoom in to the *ter* region. The peak ranges for the wild type and mutants are marked. The inset shows the the *dif* site with a single-nucleotide resolution. (C) Fold change between the coverage of the *ter* region in the mutants compared to the wild type.

membrane vesicles from *P. aeruginosa* biofilms were not treated with DNase prior to isolating DNA (30). In those vesicles, also mRNA was found and sequenced. Transcripts of the SOS response were over-expressed relative to stationary culture cells, while in the DNA, we found a coverage gradient along the *ori–ter* axis, indicating that the DNA originated from cells lysed while actively replicating. In summary, while remnants of DNA originating from outside the vesicles cannot be completely excluded, there is a strong indication that it is really the DNA inside the vesicles that is enriched for the *dif* site.

## Roles of XerCD recombinases in over-replication repair

The site-specific recombinases XerC and XerD resolve chromosome dimers at the last step of cell division and are required by all bacteria with circular chromosomes. They were detected in 641 organisms from 16 phyla (33, 47, 48). When both replication forks of circular chromosomes meet at *ter*, they collide with the divisome complex. Chromosome dimers, resulting from illegitimate recombination between left and right replichores in a fraction of the population, are resolved by the FtsK-activated XerC/XerD enzymes (49). The two replication forks often do not collide exactly at *ter*, because the left and right replichores can progress with different speeds, resulting in over-replication of DNA— including *dif*—around *ter* (50–53). The DNA enriched in OMVs might, therefore, originate from over-replication repair. In our previous work, it had to remain open if the XerCD enzymes themselves influence the composition of OMV DNA, which would imply that

they have a second role beyond dimer resolution, or if other enzymes (51) are involved as well.

Our data show that the enrichment of the *ter* region in the DNA of *E. coli* OMVs peaks exactly at *dif*. This site, i.e., the recognition sequence for the XerCD recombinases, thus, may act as an anchoring point for over-replication repair. When either *xer*C or *xer*D is deleted, the enrichment of the *ter* region becomes broader, i.e., the length of excised DNA fragments around *ter* found inside the OMVs is increased. This could imply that over-replication repair still occurs, but with reduced efficiency. Moreover, the peak at *dif* itself is strongly reduced if either *xer*C or *xer*D is deleted. Thus, the activity of these enzymes influences the composition of the DNA in OMVs, although the Ftsk–XerCD complex for the chromosome dimer resolution cannot be formed. Both recombinases can also function independently, as long as their recognition sequence is provided. They were used for the construction of markerless gene deletions (54, 55) and are exploited by phages and plasmids for integration into the chromosome (47, 56), and in some bacteria, only one recombinase is required (57). Both XerC and XerD can efficiently mediate recombination independently as shown by reporter plasmids carrying tandem *dif* sites (58). We propose that in the functionally impaired Δ*xerC* and Δ*xerD* mutants, over-replication has become more likely and is to a lesser extent resolved directly at *dif*. Possible mechanisms might involve delayed recruitment of the chromosome segregation machinery to *dif* (59) or impaired interaction with either the RecBCD enzymes required for excision of over-replicated regions (51) or the Tus proteins acting as barriers against over-replication (60).

To conclude, we show that the enrichment of the *ter* region of the bacterial chromosome in OMVs is not restricted to *D. shibae* but also found in diverse genera represented by *P. marinus*, *V. cholerae*, *E. coli,* and even biofilms of *P. aeruginosa*. The site-specific recombinases XerC and XerD are essential for the enrichment of their recognition sequence *dif* in the lumen of OMVs of *E. coli*. Given their almost universal presence in Gram-negative bacteria (33) and the strong conservation of the cell division molecular machinery, it would be interesting to unravel the underlying mechanisms in more detail.

## AUTHOR AFFILIATIONS

[1]Institute of Microbiology, Technical University of Braunschweig, Braunschweig, Germany
[2]Laboratory of Anoxygenic Phototrophs, Institute of Microbiology of the Czech Academy of Science–Centre Algatech, Třeboň, Czech Republic

## AUTHOR ORCIDs

Jürgen Tomasch  http://orcid.org/0000-0002-3914-2781

## FUNDING

| Funder | Grant(s) | Author(s) |
| --- | --- | --- |
| Deutsche Forschungsgemeinschaft | TRR51 | Irene Wagner-Döbler |
| Czech Science Foundation | GX19-28778X | Jürgen Tomasch |

## AUTHOR CONTRIBUTIONS

Johannes Mansky, Formal analysis, Investigation, Writing – review and editing | Hui Wang, Investigation | Irene Wagner-Döbler, Conceptualization, Formal analysis, funding acquisition, Supervision, Writing – original draft | Jürgen Tomasch, Conceptualization, Formal analysis, Supervision, Visualization, Writing – original draft

## DATA AVAILABILITY

The newly generated sequencing data for two to three replicate samples per *E. coli* strain were deposited at the European Nucleotide Archive (ENA; https://www.ebi.ac.uk/ena)

under accession number PRJEB62439. Accession numbers for the publically available data sets are provided in Table 1.

## ADDITIONAL FILES

The following material is available online.

## Open Peer Review

**PEER REVIEW HISTORY (review-history.pdf).** An accounting of the reviewer comments and feedback.

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
