## [Reviewer comments · Microbiology Spectrum]

Microbiology Spectrum

The Effect of site specific Recombinases XerCD on the Removal of over-replicated chromosomal DNA through Outer Membrane Vesicles in Bacteria

Johannes Mansky, Hui Wang, Irene Wagner-Döbler, and Jürgen Tomasch

Corresponding Author(s): Jürgen Tomasch, Institute of Microbiology, CAS Centre Algatech

Review Timeline:

Submission Date:	June 5, 2023
Editorial Decision:	June 9, 2023
Revision Received:	August 16, 2023
Editorial Decision:	September 10, 2023
Revision Received:	November 21, 2023
Editorial Decision:	December 18, 2023
Revision Received:	December 19, 2023
Accepted:	January 22, 2024

Editor: Silvia Cardona

Reviewer(s): The reviewers have opted to remain anonymous.

Transaction Report:

DOI: <https://doi.org/10.1128/spectrum.02343-23>

June 9, 2023

Dr. Jürgen Tomasch
Institute of Microbiology, CAS Centre Algatech
Novohradská 237
Třeboň 37901
Czech Republic

Re: Spectrum02343-23 (Role of XerCD in release of over-replicated DNA through Outer Membrane Vesicles in *Escherichia coli*)

Dear Dr. Jürgen Tomasch:

Thank you for submitting your manuscript to Microbiology Spectrum. I have read your manuscript and decided to return it for modifications without further review. The reasons for my decisions are listed below. I hope by addressing my concerns, the manuscript will have a better chance to go to the review process without risking rejection.

Major concerns:

- 1) The article proposes a conserved mechanism based on similar results from a previously analyzed bacterial species to the one shown in this article (*E. coli*). This suggestion can be further supported, given that the methodology is in place. I suggest performing a similar analysis in at least two more bacteria of different lineages. As it stands the work is too preliminary for sending out to review.
- 2) A phenotypic characterization of a mutant is not enough evidence to support a link between a gene and a phenotype. Complementation analysis is required to infer any gene-to-function link.
- 3) Please ensure that the methodology is described in the manuscript. The statement "Details on all methods, including DNA isolation, can be found in [4]" is not acceptable.
- 4) The coverage analysis shown in Fig. 2 is different from the one referenced in (4) and is not described in the manuscript. Log₂ coverage, fold change etc. Please ensure that the methodology for data analysis and the statistical significance of the fold changes are reported.

When submitting the revised version of your paper, please provide (1) point-by-point responses to the issues raised by the editor as file type "Response to editor," not in your cover letter, and (2) a PDF file that indicates the changes from the original submission (by highlighting or underlining the changes) as file type "Marked Up Manuscript - For Review Only". Please use this link to submit your revised manuscript - we strongly recommend that you submit your paper within the next 60 days or reach out to me. Detailed instructions on submitting your revised paper are below.

Link Not Available

Sincerely,

Silvia Cardona

Journals Department
Reviewer comments:

Staff Comments:

Preparing Revision Guidelines

Please return the manuscript within 60 days; if you cannot complete the modification within this time period, please contact me. If you do not wish to modify the manuscript and prefer to submit it to another journal, please notify me of your decision immediately so that the manuscript may be formally withdrawn from consideration by Microbiology Spectrum.

Response to the raised concerns:

- 1) The article proposes a conserved mechanism based on similar results from a previously analyzed bacterial species to the one shown in this article (*E. coli*). This suggestion can be further supported, given that the methodology is in place. I suggest performing a similar analysis in at least two more bacteria of different lineages. As it stands the work is too preliminary for sending out to review.

We agree that for the proposal of a general mechanism proven from more bacteria is needed. As we cannot perform the experiments by ourselves anymore we searched the literature for other sequencing studies of DNA from OMVs. We identified three studies. For Cyanobacterium *Prochlorococcus marinus* and both chromosomes of Gammaproteobacterium *Vibrio cholerae*, we could clearly identify peaks at the *dif* sites as we found for *D. shibae* and *E. coli*. The OMVs isolated from Gammaproteobacterium *Pseudomonas aeruginosa* were predominantly formed during explosive cell lysis. However, we also could identify a peak at *dif* site that might have originated from overreplication repair. Thus, studies from other organisms support our hypothesis.

- 2) A phenotypic characterization of a mutant is not enough evidence to support a link between a gene and a phenotype. Complementation analysis is required to infer any gene-to-function link.

We fully agree that complemented strains would be needed for a full support of the proposed link. We hadn't finished the strains yet when the lab closed. However, both independent *xerC* and *xerD* mutants showing the same phenotype, provides already limited evidence for the proposed hypothesis. We made it now clear that these data are preliminary and have to be treated with caution.

- 3) Please ensure that the methodology is described in the manuscript. The statement "Details on all methods, including DNA isolation, can be found in [4] is not acceptable.

We submitted the study as an observation with very limited space. We felt that the required shortening of the methods section would provide no benefit to the reader over the methods section of our first OMV paper that we cite here and that is very detailed. We are convinced it will be a reliable guide to other researchers.

- 3) The coverage analysis shown in Fig. 2 is different from the one referenced in (4) and is not described in the manuscript. Log₂ coverage, fold change etc. Please ensure that the methodology for data analysis and the statistical significance of the fold changes are reported.

We are sorry for the missing information that we added now to the methods section.

September 10, 2023

Dr. Jürgen Tomasch
Institute of Microbiology, CAS Centre Algatech
Novohradská 237
Třeboň 37901
Czech Republic

Re: Spectrum02343-23R1 (Release of over-replicated DNA through Outer Membrane Vesicles in Bacteria)

Dear Dr. Jürgen Tomasch:

Thank you for submitting your manuscript to Microbiology Spectrum. Your article has been reviewed by two experts in the field. Both reviewers found the findings interesting and have provide comments that, if addressed, will enhance the quality of the work substantially. In their notes, reviewers noted one of the conclusions is not supported by the data presented. However, they do not request new experiments, but to rephrase the conclusions, accordingly. Also, one reviewer recommends to modify the title to reflect the main topic, which is the effect of XerCD on the removal of over-replicated chromosomal DNA. Finally, one reviewer is unsure if the type of article chosen for this work (observation) is suitable for presenting the data with rigour. This reviewer has recommendations to address this issue. Their detailed recommendations are provided below.

Link Not Available

Sincerely,

Silvia Cardona

Journals Department
Reviewer comments:

Reviewer #1 (Comments for the Author):

Line 1:

The title does not represent the data covered in the research. The release of DNA through OMVs is not the topic of the experiments. The effect of XerCD on the removal of over-replicated chromosomal DNA is the main focus of the experiments presented.

Lines 24-26, "Our data suggest a conserved mechanism for repair and removal of over-replicated DNA through outer membrane vesicles and an active role of the site-specific XerCD recombinase complex therein." and

Lines 134-135: "The site-specific recombinases XerC and XerD might play an active role for enrichment of their recognition sequence dif in the lumen of OMVs."

My interpretation of these sentences is that XerCD is actively involved in the translocation of DNA to the lumen of OMVs. The data presented here do not cover the removal of over-replicated DNA from the cell through OMVs. The issue of how the excised DNA ends up in the OMVs is not studied here. The OMVs are used as a source of the DNA and assumed to contain a representative sample of the excised DNA. Maybe the authors can rephrase these sentences and be more specific on the role of XerCD.

Lines 57-59: "Are the XerCD enzymes directly involved in the excision of over-replicated DNA fragments around ter, implicating that they might have a second function in addition to resolving chromosome dimers?"

The experiments presented in this manuscript do not address whether the XerCD enzymes are directly or indirectly involved in this process. Indirect effects can not be excluded and I would therefore suggest to remove the word "directly" and rephrase the second half of the sentence to reflect the fact that the observed effects might be an indirect result of the absence of one of the enzymes.

Lines 59-60: "Therefore, we investigated the DNA composition in the lumen of OMVs produced by deletion mutants of XerC and XerD in *E. coli*."

Deletion mutants will not answer this question, they will just answer if the enzymes (or actually the genes) are essential for this process.

Since it is the genes that are deleted I would refer to xerC and xerD rather than XerC and XerD.

Line 74:

This is a reference to a reference. Please refer to *Genome Biol Evol* 10:359-369. <https://doi.org/10.1093/gbe/evy005> directly.

Lines 103 -104: "... then the mutants apparently had more waste to get rid of."

It is intriguing that at time 0 the number of vesicles per cell are the same (~10 vesicles per cell) for the three strains, indicating that during preculturing conditions the vesicle formation was likely similar. But during the 12 h cultivation the wild-type cells hardly produce vesicles while the xerC and xerD mutant strains continue vesicle production. This means that the vesicles after 12 h cultivation were produced during a different stage of culturing. For the wild-type a major part of the vesicles were produced before the 12 h cultivation, while >95% (rough calculation, going from 2×10^8 to 5×10^9 vesicles/mL) of the vesicles obtained from the mutant strains were produced during the 12 h cultivation.

To be able to claim that the mutants had more waste to get rid of you would need to show that the newly produced vesicles contain DNA.

Another question is whether the vesicles are produced as a means to get rid of the excess DNA (active process, as is suggested by the phrase "had more waste to get rid of") or if the DNA just happens to be incorporated in the vesicles (passive process). Maybe the authors can rephrase this sentence.

Lines 116-117: "These data suggest impaired binding or functioning of the XerCD recombinase complex at the dif site when one of the homologs is knocked out."

In the deletion mutants there is no XerCD recombinase complex.

Please rephrase this sentence.

Suggestions regarding potential typos:

Line 80: Supplementary

Line 91: remainder

Line 134: is

Reviewer #2 (Comments for the Author):

Overall comments:

The authors' condensed resubmitted article describes reanalyzed and newly collected DNA sequencing data from bacterial OMV isolations. Some proteobacterial species they have explored in this and prior studies was examined and appears to show that some OMVs are enriched with chromosomal DNA regions (dif and ter) specifically over-generated during cell division. The study

is provocative and likely impactful, but the article's current very condensed/ shortened format makes the findings and methods briefly described a difficult and at times confusing to follow in places. Many questions related to how methods were compared (eg. were vesicle particles/ml standardized for DNA extraction) and other questions about the aim of the study are posed in the comments below to help the authors enhance a reader's understanding of the content (as they didn't have the benefit of the longer marked up manuscript version). The article would also benefit from more citations to prove arguments/ statements made, as well as additional proofreading and grammar editing; many grammar editing programs will catch many of these sentence structure/grammar issues that weren't noted in comments but present in the manuscript.

Specific comments:

ABSTRACT:

Line 12. This statement about OMVs being produced by all prokaryotes is incorrect as written; not all prokaryotes have outer membranes. Please correct this statement (as on lines 28-29) or omit it. A similar oversimplification of prokaryotes occurs in the author's conclusions statement made on lines 135-136 as well.

Line 18-20. This statement regarding E.coli deletion mutants lacks a clear interpretation or explanation of outcomes. Please consider rephrasing findings here to interpret the outcomes of results for the reader as few read beyond the abstract.

IMPORTANCE:

Lines 22-26. The authors do not prove or show evidence of how *ter* is enriched in the species OMVs they DNA sequenced. This section should be carefully revised to highlight why *dif* and *ter* DNA enrichment in OMVs is important in Gram-negative species OMV formation. Explain to a reader why this is potentially important to study/ understand.

INTRODUCTION:

Line 30 "...vesicles of Gram negative bacteria (OMV) have often been found to contain DNA,..." Can the authors clarify the following in this statement: i) What articles are being referred to here, because reference 3 does not sufficiently address this content. ii) Is the DNA referred to here actually confirmed to be located within the OMV or was it just carried over with the isolated vesicles (artifactual)? How common is DNA containment in vesicles? As this is a major rationale for the study these could be more clearly justified/ explained in this paragraph.

Line 34: "Based on multiple evidence..." Please cite this evidence here (either reference papers and summarize main arguments) as it was not clear from the first statement about prior alphaproteobacterial work.

Line 35 "...vesicles, which were ejected from the dividing cell's division plane,.." Please discuss this evidence in more detail as it is again a major argument poorly described in the hypothesis but not well argued in the introduction.

Line 45: "The DNA enriched in OMV might therefore originate..." Grammar issues related to "therefore usage and placement. Also the authors should briefly describe what is known about the DNA enriched in OMVs? Are DNA verified to be inside the vesicles (see carryover question above)? DNA may be attracted to the surface of the vesicles drawn by Mg²⁺ and Ca²⁺ that associate with outer layer OMV lipopolysaccharides.

Lines 46-50. The aim and arguments posed to rationalize the study, were a bit confusing. Do the authors seek to clarify why over-replication by XerCD occurs? It was not clear how monitoring OMV DNA content is relates and was a logic gap based on the introductory content provided. What are the authors trying to prove, unequal XerCD replication or the fact that these over-replicated DNA products are dealt with by cells through their removal in OMVs during cell division? The study does not really explain how that happens as alluded to in the "Importance section". Were mid-log cells selected for this study versus the more common stationary phase cells to enrich cell replication-derived OMVs versus stationary; this was not clear in the methods?

Line 49-50. More effort needs to be made before this statement and in the methods to explain/ prove how DNA was extracted from within OMVs for sequencing (and not also contaminating carryover with vesicles) or from other descriptions of explosive cell lysis in lines 81-93. Were all OMVs DNA sequenced in the study collected at the same stages of culturing (mid or late log, stationary phase?). Did growth have an impact? Additionally, tangential flow filtration (tff) used by the authors to isolate OMVs could just as easily increase and concentrate released DNA along with accumulated OMVs. The authors need clarify and prove this DNA is inside or attached to the vesicle based on these methods, as this is major part of their hypothesis and main argument at many points in the paper. The hypothesis may still hold even if the DNA is on the surface of vesicles and not inside, so some effort should be made on this topic.

Line 59-60. Please explain how luminal DNA only was isolated from OMVs here. After reviewing the fluorescent staining images from cited study, the vesicles shown with FM1-43 lipid stained and DAPI stained OMVs did not always show that DAPI-stained DNA was solely located within OMV lumen (Fig 1G-H) of the cited reference #4. How much DAPI signal was in the background of the image and was the image background corrected? Was this cited study (reference 4) the only evidence of luminal DNA in OMVs to prove the author's arguments?

Line 85. "It is therefore expected that it might contain also DNA other than dif." Hard to follow statement; many grammar issues hindering meaning. This type of situation occurs often in this short article. Please carefully edit the article for grammar.

Line 81 (Paragraph). This whole section is missing DNA GenBank accession numbers and reference citations for the DNA described and used in the study. The authors should make more effort to properly cite prior collected data used in the study. Figure 1 and 2 of the author's study should provide an added chart or quantification of the total sequenced DNA reads, where the % of ter and dif identified in coverage is summarized with respect to other possible identified or unknown DNA sequences recovered from the OMVs. How much of the OMV DNA recovered (% of total) is actually ter or dif DNA as opposed to other random DNA fragments?

Line 94-104. Given that XerCD mutants produced very different amounts of vesicles per ml amounts, did DNA extractions from all OMV preps account for the differences in vesicles per ml? This could significantly skew the arguments made in the next paragraph which state fold change increases and decreases in ter and dif DNA content. But, this may actually be due to reductions in the isolated particles/ml of OMVs recovered by these mutant OMVs. Can the authors comment on this?

Line 105: is this paragraph continuing arguments from the paragraph above? It is not clear what results are being discussed? Assume OMV DNA from XerC/D mutants? Unclear and hard to follow.

Lines 119-122: Unclear wording; what does "The peak of DNA in OMVs" really refer to here? What peak? The statement in lines 120-122 is very hard to follow as written. What is meant or actually referred to by "independent mutants" here?

Line 123: Sentence fragment. "The XerCD recombination complex is highly adaptable." Adaptable to with respect to what?

Line 128: Assume E. coli gene deletion mutants are meant here in the statement "...functionally impaired Δ xerC and Δ xerD mutants..." ?

Line 134: Please clearly state other species in the study ter region enrichment also occurs in in this statement.

Staff Comments:

Preparing Revision Guidelines

Please return the manuscript within 60 days; if you cannot complete the modification within this time period, please contact me. If you do not wish to modify the manuscript and prefer to submit it to another journal, please notify me of your decision immediately so that the manuscript may be formally withdrawn from consideration by Microbiology Spectrum.

While we are willing to consider a revised version of this paper at Spectrum, it would be in your best interest to improve the writing. I recommend that you ask a colleague of yours who is a native English speaker to read and provide you some feedback on the writing. You are also welcome to use one of the services here: <https://journals.asm.org/content/language-editing-services>

Line 1:

The title does not represent the data covered in the research. The release of DNA through OMVs is not the topic of the experiments. The effect of XerCD on the removal of over-replicated chromosomal DNA is the main focus of the experiments presented.

Lines 24-26: “Our data suggest a conserved mechanism for repair and removal of over-replicated DNA through outer membrane vesicles and an active role of the site-specific XerCD recombinase complex therein.” and

Lines 134-135: “The site-specific recombinases XerC and XerD might play an active role for enrichment of their recognition sequence *dif* in the lumen of OMVs.”

My interpretation of these sentences is that XerCD is actively involved in the translocation of DNA to the lumen of OMVs. The data presented here do not cover the removal of over-replicated DNA from the cell through OMVs. The issue of how the excised DNA ends up in the OMVs is not studied here. The OMVs are used as a source of the DNA and assumed to contain a representative sample of the excised DNA. Maybe the authors can rephrase these sentences and be more specific on the role of XerCD.

Lines 57-59: “Are the XerCD enzymes directly involved in the excision of over-replicated DNA fragments around *ter*, implicating that they might have a second function in addition to resolving chromosome dimers?”

The experiments presented in this manuscript do not address whether the XerCD enzymes are directly or indirectly involved in this process. Indirect effects can not be excluded and I would therefore suggest to remove the word “directly” and rephrase the second half of the sentence to reflect the fact that the observed effects might be an indirect result of the absence of one of the enzymes.

Lines 59-60: “Therefore, we investigated the DNA composition in the lumen of OMVs produced by deletion mutants of XerC and XerD in *E. coli*.”

Deletion mutants will not answer this question, they will just answer if the enzymes (or actually the genes) are essential for this process.

Since it is the genes that are deleted I would refer to *xerC* and *xerD* rather than XerC and XerD.

Line 74:

This is a reference to a reference. Please refer to Genome Biol Evol 10:359–369. <https://doi.org/10.1093/gbe/evy005> directly.

Lines 103 -104: “... then the mutants apparently had more waste to get rid of.”

It is intriguing that at time 0 the number of vesicles per cell are the same (~10 vesicles per cell) for the three strains, indicating that during preculturing conditions the vesicle formation was likely similar. But during the 12 h cultivation the wild-type cells hardly produce vesicles while the *xerC* and *xerD* mutant strains continue vesicle production. This means that the vesicles after 12 h cultivation were produced during a different stage of culturing. For the wild-type a major part of the vesicles were produced before the 12 h cultivation, while >95% (rough calculation, going from 2×10^8 to 5×10^9 vesicles/mL) of the vesicles obtained from the mutant strains were produced during the 12 h cultivation.

To be able to claim that the mutants had more waste to get rid of you would need to show that the newly produced vesicles contain DNA.

Another question is whether the vesicles are produced as a means to get rid of the excess DNA (active process, as is suggested by the phrase "had more waste to get rid of") or if the DNA just happens to be incorporated in the vesicles (passive process).

Maybe the authors can rephrase this sentence.

Lines 116-117: "These data suggest impaired binding or functioning of the XerCD recombinase complex at the *dif* site when one of the homologs is knocked out."

In the deletion mutants there is no XerCD recombinase complex.

Please rephrase this sentence.

Suggestions regarding potential typos:

Line 80: Supplementury

Line 91: remainder

Line 134: is

Reviewer comments:

Reviewer #1 (Comments for the Author):

Line 1:

The title does not represent the data covered in the research. The release of DNA through OMVs is not the topic of the experiments. The effect of XerCD on the removal of over-replicated chromosomal DNA is the main focus of the experiments presented.

Answer: Title was changed as suggested.

Lines 24-26, "Our data suggest a conserved mechanism for repair and removal of over-replicated DNA through outer membrane vesicles and an active role of the site-specific XerCD recombinase complex therein." and

Lines 134-135: "The site-specific recombinases XerC and XerD might play an active role for enrichment of their recognition sequence *dif* in the lumen of OMVs."

My interpretation of these sentences is that XerCD is actively involved in the translocation of DNA to the lumen of OMVs. The data presented here do not cover the removal of over-replicated DNA from the cell through OMVs. The issue of how the excised DNA ends up in the OMVs is not studied here. The OMVs are used as a source of the DNA and assumed to contain a representative sample of the excised DNA. Maybe the authors can rephrase these sentences and be more specific on the role of XerCD.

Answer: Thank you for your comments. We have rephrased the "Importance" paragraph (formerly lines 24-26) in the following way:

"Our data show that OMVs from diverse Proteobacteria are enriched for over-replicated sequences around the terminus of replication *ter* and the site-specific XerCD recombinases influence this enrichment. Clearing the divisome from over-replicated parts of the bacterial chromosome might be a novel function of OMVs."

Regarding lines 134-135: Of course we did not want to imply that the XerCD recombinases are actively involved in the translocation of DNA into the lumen of OMVs. We showed, however, that the enrichment of *ter* and *dif* changes when either *xerC* or *xerD* is deleted, meaning the activity of these enzymes is essential for the enrichment. We have rephrased the statement in the following way:

"To conclude, we show that the enrichment of the *ter* region of the bacterial chromosome in OMVs is not restricted to *D. shibae*. The site-specific recombinases XerC and XerD are essential for enrichment of their recognition sequence *dif* in the lumen of OMVs. Given their almost universal presence in Bacteria [36] and the strong conservation of the cell division molecular machinery it would be interesting to unravel the underlying mechanisms in more detail."

Lines 57-59: "Are the XerCD enzymes directly involved in the excision of over-replicated DNA fragments around *ter*, implicating that they might have a second function in addition to resolving chromosome dimers?"

The experiments presented in this manuscript do not address whether the XerCD enzymes are directly or indirectly involved in this process. Indirect effects can not be excluded and I would therefore suggest to remove the word "directly" and rephrase the second half of the sentence to reflect the fact that the observed effects might be an indirect result of the absence of one of the enzymes.

Answer: We have carefully considered your comments above and elsewhere and rephrased the sentence in the following way:

"Deletion of *xerC* or *xerD* in *E. coli* reduces the enrichment peak directly at the *dif* sequence, while the enriched DNA region around *ter* becomes broader, demonstrating that either enzyme influences the contents of DNA inside the lumen of OMVs. We hypothesize that the intra-vesicle DNA is the result of

over-replication repair and the XerCD enzymes contribute to this, providing them with a new function in addition to resolving chromosome dimers."

Lines 59-60: "Therefore, we investigated the DNA composition in the lumen of OMVs produced by deletion mutants of XerC and XerD in *E. coli*."

Deletion mutants will not answer this question, they will just answer if the enzymes (or actually the genes) are essential for this process.

Since it is the genes that are deleted I would refer to *xerC* and *xerD* rather than XerC and XerD.

Answer: Was rephrased as suggested. Sentence now reads:

"(2) Are the XerCD enzymes essential for the enrichment of *ter* and *dif* in the DNA inside OMVs? When these enzymes are resolving chromosome dimers, no fragments containing *dif* are produced. Therefore, we investigated the DNA composition in the lumen of OMVs produced by deletion mutants of *xerC* and *xerD* in *E. coli*."

Line 74:

This is a reference to a reference. Please refer to Genome Biol Evol 10:359-369.

<https://doi.org/10.1093/gbe/evy005> directly.

Answer: Thank you, we added the correct reference. In addition, we also uploaded the used scripts to github. The link is provided in the material and methods section.

Lines 103 -104: "... then the mutants apparently had more waste to get rid of."

It is intriguing that at time 0 the number of vesicles per cell are the same (~10 vesicles per cell) for the three strains, indicating that during preculturing conditions the vesicle formation was likely similar. But during the 12 h cultivation the wild-type cells hardly produce vesicles while the *xerC* and *xerD* mutant strains continue vesicle production. This means that the vesicles after 12 h cultivation were produced during a different stage of culturing. For the wild-type a major part of the vesicles were produced before the 12 h cultivation, while >95% (rough calculation, going from 2×10^8 to 5×10^9 vesicles/mL) of the vesicles obtained from the mutant strains were produced during the 12 h cultivation.

To be able to claim that the mutants had more waste to get rid of you would need to show that the newly produced vesicles contain DNA.

Another question is whether the vesicles are produced as a means to get rid of the excess DNA (active process, as is suggested by the phrase "had more waste to get rid of") or if the DNA just happens to be incorporated in the vesicles (passive process).

Maybe the authors can rephrase this sentence.

Answer: Your careful observations are really interesting. However, we do not know the half-life of the produced vesicles. The samples from 12 h might contain none or all or a certain fraction of the vesicles produced at 1 h. What we do see, though, is that the number of vesicles per cell is similar between wt and mutants during the first 2 h of cultivation. Then, at 4 h, it declines in the wt and remains high in the mutants. We think it is possible to conclude that the mutant cultures produced more vesicles per cell, particularly during stationary phase.

Regarding the passive or active incorporation of the DNA into the vesicles: Since we do not know the mechanism, we rephrased the sentence accordingly. Paragraph now reads:

"The ratio of vesicles per cell was similar for wild-type and mutants during the first 2 h of growth. Then, at 4 h, it dropped to 0.6 - 0.2 for the wild-type while it remained between 3 and 10 for both mutants (Supplementary Figure). If our hypothesis is true and the DNA in OMVs represents over-replicated fragments, then more such waste was produced in the mutant OMVs."

Lines 116-117: "These data suggest impaired binding or functioning of the XerCD recombinase complex at the *dif* site when one of the homologs is knocked out."

In the deletion mutants there is no XerCD recombinase complex.

Please rephrase this sentence.

Answer: Right. Thank you. Sentence now reads:

“Since the XerCD-FtsK-complex cannot be formed when either *xerC* or *xerD* are knocked out, these data reflect the activity of the remaining recombinase homolog.”

Suggestions regarding potential typos:

Line 80: Supplementary

Line 91: remainder

Line 134: is

Answer: Thank you, was corrected or reformulated.

Reviewer #2 (Comments for the Author):

Overall comments:

The authors' condensed resubmitted article describes reanalyzed and newly collected DNA sequencing data from bacterial OMV isolations. Some proteobacterial species they have explored in this and prior studies was examined and appears to show that some OMVs are enriched with chromosomal DNA regions (*dif* and *ter*) specifically over-generated during cell division. The study is provocative and likely impactful, but the article's current very condensed/ shortened format makes the findings and methods briefly described a difficult and at times confusing to follow in places. Many questions related to how methods were compared (eg. were vesicle particles/ml standardized for DNA extraction) and other questions about the aim of the study are posed in the comments below to help the authors enhance a reader's understanding of the content (as they didn't have the benefit of the longer marked up manuscript version). The article would also benefit from more citations to prove arguments/ statements made, as well as additional proofreading and grammar editing; many grammar editing programs will catch many of these sentence structure/grammar issues that weren't noted in comments but present in the manuscript.

Answer: Thank you! We appreciate the valuable criticism.

Specific comments:

ABSTRACT:

Line 12. This statement about OMVs being produced by all prokaryotes is incorrect as written; not all prokaryotes have outer membranes. Please correct this statement (as on lines 28-29) or omit it. A similar oversimplification of prokaryotes occurs in the author's conclusions statement made on lines 135-136 as well.

Answer: Of course, sorry for the mistake. We replaced “prokaryotes” by “Gram negative Bacteria” as only those have outer membranes. In the Introduction we generalized that membrane vesicles are produced by cells from all domains of life.

Line 18-20. This statement regarding *E.coli* deletion mutants lacks a clear interpretation or explanation of outcomes. Please consider rephrasing findings here to interpret the outcomes of results for the reader as few read beyond the abstract.

Answer: We have added an interpretation. Sentence now reads:

Deletion of *xerC* or *xerD* in *E. coli* reduced the enrichment peak directly at the *dif* sequence, while the enriched DNA region around *ter* becomes broader, demonstrating that either enzyme influences the contents of DNA inside the lumen of OMVs.

IMPORTANCE:

Lines 22-26. The authors do not prove or show evidence of how *ter* is enriched in the species OMVs they DNA sequenced. This section should be carefully revised to highlight why *dif* and *ter* DNA enrichment in OMVs is important in Gram-negative species OMV formation. Explain to a reader why this is potentially important to study/ understand.

Answer: We have reformulated this paragraph, requested also by reviewer #1. It now reads: Imprecise termination of replication can lead to over-replicated parts of bacterial chromosomes that have to be excised and removed from the dividing cell. The underlying mechanism is poorly understood. Our data show that OMVs from diverse Proteobacteria are enriched for over-replicated sequences around the terminus of replication *ter* and the site-specific XerCD recombinases influence this enrichment. Clearing the divisome from over-replicated parts of the bacterial chromosome might be a novel function of OMVs.

INTRODUCTION:

Line 30 "...vesicles of Gram negative bacteria (OMV) have often been found to contain DNA,..." Can the authors clarify the following in this statement: i) What articles are being referred to here, because reference 3 does not sufficiently address this content. ii) Is the DNA referred to here actually confirmed to be located within the OMV or was it just carried over with the isolated vesicles (artifactual)? How common is DNA containment in vesicles? As this is a major rationale for the study these could be more clearly justified/ explained in this paragraph.

Answer: In the introduction, we cite numerous papers that found DNA inside OMVs, and extra-vesicle DNA was always digested prior to isolating intra-vesicular DNA. (Line 37 - 54).

However, the strength of DNase digestion influences the completeness of removal of extra-vesicular DNA. This point is now extensively discussed. We added a new paragraph to the discussion ("**DNA composition of OMVs**") where we also refer to our previous study of OMVs from *D. shibae*, where we sequenced the vesicle DNA with and without DNase treatment. (Line 196 - 212)

Line 34: "Based on multiple evidence..." Please cite this evidence here (either reference papers and summarize main arguments) as it was not clear from the first statement about prior alphaproteobacterial work.

Answer: The evidence from our previous work was meant, which was based on time-lapse microscopy, proteome, transcriptome, and metabolome data. We describe these results in more detail now. (Line 65 - 81).

Line 35 "...vesicles, which were ejected from the dividing cell's division plane,..." Please discuss this evidence in more detail as it is again a major argument poorly described in the hypothesis but not well argued in the introduction.

Answer: See comment above.

Line 45: "The DNA enriched in OMV might therefore originate..." Grammar issues related to "therefore usage and placement. Also the authors should briefly describe what is known about the DNA enriched in OMVs? Are DNA verified to be inside the vesicles (see carryover question above)? DNA may be attracted to the surface of the vesicles drawn by Mg²⁺ and Ca²⁺ that associate with outer layer OMV lipopolysaccharides.

Answer: We have re-formulated statements in the Abstract and Importance paragraphs. The issue of intra- versus extra-vesicle DNA from OMVs is addressed in the Introduction and Discussion, including our previous work where we sequenced both preparations of vesicle DNA.

Lines 46-50. The aim and arguments posed to rationalize the study, were a bit confusing. Do the authors seek to clarify why over-replication by XerCD occurs? It was not clear how monitoring OMV DNA content is relates and was a logic gap based on the introductory content provided. What are the authors trying to prove, unequal XerCD replication or the fact that these over-replicated DNA products are dealt with by cells through their removal in OMVs during cell division? The study does not really explain how that happens as alluded to in the "Importance section". Were mid-log cells selected for

this study versus the more common stationary phase cells to enrich cell replication-derived OMVs versus stationary; this was not clear in the methods?

Answer: Those are a lot of questions! The rationale of the introduction is the following:

- (1) DNA has routinely been found inside OMVs. Many references are now provided. (Line 35 - 54).
- (2) Given the mechanism of OMV biogenesis by blebbing from the outer membrane, it is unclear how DNA can be incorporated into OMVs. (Line 55 - 64).
- (3) Our hypothesis based on previous work with *D. shibae*: (Line 65 - 81).

Then, we continue to explain which part of this hypothesis is explored in the current paper.

Regarding your suggestion to compare the DNA content of exponential to stationary phase OMVs: It is certainly a good idea. Here (and in our previous work) we harvested OMVs at the end of the exponential growth phase.

Line 49-50. More effort needs to be made before this statement and in the methods to explain/ prove how DNA was extracted from within OMVs for sequencing (and not also contaminating carryover with vesicles) or from other descriptions of explosive cell lysis in lines 81-93. Were all OMVs DNA sequenced in the study collected at the same stages of culturing (mid or late log, stationary phase?). Did growth have an impact? Additionally, tangential flow filtration (tff) used by the authors to isolate OMVs could just as easily increase and concentrate released DNA along with accumulated OMVs. The authors need clarify and prove this DNA is inside or attached to the vesicle based on these methods, as this is major part of their hypothesis and main argument at many points in the paper. The hypothesis may still hold even if the DNA is on the surface of vesicles and not inside, so some effort should be made on this topic.

Answer: Thank you for this comment. We dedicate now one section to the discussion of the vesicle DNA content:

“For *P. marinus*, *D. shibae*, *V. cholerae* and the newly analyzed *E. coli* strains, the vesicles were treated by DNase prior to analyzing the DNA inside the vesicle lumen. However, the effectiveness of DNase treatment plays a large role in the enrichment of protected DNA and a complete removal of extra-vesicular DNA cannot be guaranteed [39]. For *D. shibae* we previously sequenced DNA from both, DNase treated and untreated vesicle enrichments and could show that digestion of unprotected DNA results in a reduction of reads mapping outside the *ter* region. In case of *V. cholerae*, the sampling time point was chosen to minimize DNA originating from lysed cells and two consecutive digestion steps were performed. Besides enrichment of the *ter* region, some other short specific regions and in particular phage DNA was found to be overrepresented. DNA within a phage is also shielded from DNase activity. The Membrane vesicles from *P. aeruginosa* biofilms were not treated by DNase prior to isolating DNA [31]. In those vesicles, also mRNA was found and sequenced. Transcripts of the SOS response were over-expressed relative to stationary culture cells, while in the DNA we found a coverage gradient along the *ori-ter* axis, indicating that the DNA originates from cells lysed while actively replicating. In summary, while remnants of DNA originating from outside the vesicles cannot be completely excluded, there is strong indication that it is really the DNA inside the vesicles that is enriched for the *dif* site.”

Line 59-60. Please explain how luminal DNA only was isolated from OMVs here. After reviewing the fluorescent staining images from cited study, the vesicles shown with FM1-43 lipid stained and DAPI stained OMVs did not always show that DAPI-stained DNA was solely located within OMV lumen (Fig 1G-H) of the cited reference #4. How much DAPI signal was in the background of the image and was the image background corrected? Was this cited study (reference 4) the only evidence of luminal DNA in OMVs to prove the author's arguments?

Answer: We expanded the methods section to provide a detailed description of the vesicle isolation and removal of external DNA. Indeed, the fluorescence micrographs of *D. shibae* OMVs showed rare cases where the DAPI signal might originate from the presence of DNA attached to the vesicles (there was almost no background-signal and no subtraction had been performed). However, when we developed the method further, we sequenced DNA from both, DNase treated and untreated vesicle enrichments and could show that digestion of unprotected DNA results in a reduction of reads mapping outside the *ter* region. Although we cannot exclude remaining residual external DNA, the observed enrichment of mapped reads clearly shows that DNase treatment was successful. There are numerous studies that showed DNA within vesicles, although for most of them no sequencing data exist. We expanded the introduction accordingly.

Line 85. "It is therefore expected that it might contain also DNA other than dif." Hard to follow statement; many grammar issues hindering meaning. This type of situation occurs often in this short article. Please carefully edit the article for grammar.

Answer: We modified this part in order to make it clearer: "For *P. aeruginosa* the sequenced DNA reportedly originated mainly from OMVs formed in biofilms during explosive cell lysis. In this process the whole cellular DNA content is released and can be attached to the surface or included in the lumen of newly formed vesicles. It is therefore expected that over-replicated DNA from the last stage of cell division might not be particularly enriched."

Line 81 (Paragraph). This whole section is missing DNA GenBank accession numbers and reference citations for the DNA described and used in the study. The authors should make more effort to properly cite prior collected data used in the study. Figure 1 and 2 of the author's study should provide an added chart or quantification of the total sequenced DNA reads, where the % of *ter* and *dif* identified in coverage is summarized with respect to other possible identified or unknown DNA sequences recovered from the OMVs. How much of the OMV DNA recovered (% of total) is actually *ter* or *dif* DNA as opposed to other random DNA fragments?

Answer: Accession numbers and references had been provided in Supplementary Table S1 before. We provide this information now in Table 1 together with read mapping statistics showing an enrichment of the *ter* region in the analysed strains. As the *dif* site itself is much smaller than the read size, it is in our opinion not of much use to calculate mapping statistics based on read counts. We compared the counts mapped to the *ter* region to counts from random regions outside *ter*. As the random number generator had the tendency to produce smaller numbers within the required range, samples near the start of the DNA sequence were overrepresented. Therefore, we divided the chromosome into 10 equal parts excluding *ter* and sampled 20 times within each segment. As we have only one *ter* site (sample size = 1) compared to many randomly sampled sites, no statistical test could be applied in a meaningful way. However, we think that the enrichment of *ter* becomes clear from the test as well as graphical inspection shown in the figure.

Line 94-104. Given that XerCD mutants produced very different amounts of vesicles per ml amounts, did DNA extractions from all OMV preps account for the differences in vesicles per ml? This could significantly skew the arguments made in the next paragraph which state fold change increases and decreases in *ter* and *dif* DNA content. But, this may actually be due to reductions in the isolated particles/ml of OMVs recovered by these mutant OMVs. Can the authors comment on this?

Answer: The same amount of input DNA was used for all samples and correction for sequencing depth was performed prior to analysis. The different amount of vesicles in the enriched fraction from the mutants should therefore not influence the read composition of the sequenced samples. Furthermore, the very specific changes in both directions directly at the *dif* site are a strong indication that the observed patterns are not due to such a general effect.

Line 105: is this paragraph continuing arguments from the paragraph above? It is not clear what results are being discussed? Assume OMV DNA from XerC/D mutants? Unclear and hard to follow.

Answer: The paragraph above line 105 describes OMV secretion in *E. coli* cultures based on cell counts and vesicle counts of wild-type and *xerCD* mutants (now Figure 2), while the next paragraph describes the analysis of the DNA composition of OMVs from *E. coli* wild-type and *xerCD* mutants. We added a clarifying sentence and Table 1 quantifying the enrichment of *ter*.

This paragraph now starts with the following sentence:

For all three strains we found the *ter* region overrepresented in the DNA isolated from the OMV's lumen (Table 1).

Lines 119-122: Unclear wording; what does "The peak of DNA in OMVs" really refer to here? What peak?

The statement in lines 120-122 is very hard to follow as written. What is meant or actually referred to by "independent mutants" here?

Line 123: Sentence fragment. "The XerCD recombination complex is highly adaptable." Adaptable to with respect to what?

Line 128: Assume *E. coli* gene deletion mutants are meant here in the statement "...functionally impaired $\Delta xerC$ and $\Delta xerD$ mutants..." ?

Answer: We think some of the confusion may have been caused by the need for brevity in our previous manuscript. Since we now had the chance to write a full paper, we shifted the paragraph "Roles of XerCD recombinases in over-replication repair" to the discussion and start by explaining the functions of these enzymes and our hypothesis (line 214 - 225). Then, we re-wrote the interpretation of our findings (line 226 - 239), taking into account your comments above. This part now reads:

"Our data show that the enrichment of the *ter* region in the DNA of *E. coli* OMVs peaks exactly at *dif*. This site, i.e. the recognition sequence for the XerCD recombinases, thus may act as an anchoring point for over-replication repair. When either *xerC* or *xerD* is deleted, the enrichment of the *ter* region becomes broader, i.e. the length of excised DNA fragments around *ter* found inside the OMVs is increased. This could imply that over-replication repair still occurs, but with reduced efficiency. Moreover, the peak at *dif* itself is strongly reduced if either *xerC* or *xerD* is deleted. Thus, the activity of these enzymes influences the composition of the DNA in OMVs, although the Ftsk-XerCD complex for chromosome dimer resolution can probably not be formed. Both recombinases can also function independently, as long as their recognition sequence is provided. They were used for the construction of markerless gene deletions [54][55], are exploited by phages and plasmids for integration into the chromosome [47][56], and in some bacteria only one recombinase is required [57]. Both XerC and XerD can efficiently mediate recombination independently as shown by reporter plasmids carrying tandem *dif* sites [58]."

Line 134: Please clearly state other species in the study *ter* region enrichment also occurs in in this statement.

Answer: Thank you for reminding us. Sentence now reads:

"To conclude, we show that the enrichment of the *ter* region of the bacterial chromosome in OMVs is not restricted to *D. shibae*, but also found in such diverse genera as *P. marinus*, *V. cholerae*, *E. coli* and even biofilms of *P. aeruginosa*."

Re: Spectrum02343-23R2 (The Effect of site specific Recombinases XerCD on the Removal of over-replicated chromosomal DNA through Outer Membrane Vesicles in Bacteria)

Dear Dr. Jürgen Tomasch:

Your work has been revised and has improved substantially. Still, there are minor changes the reviewers want you to include; below, you will find their comments, instructions from the Spectrum editorial office, and the reviewer's comments.

Revision Guidelines

Sincerely,
Silvia Cardona
Editor
Microbiology Spectrum

Reviewer #1 (Comments for the Author):

I have reviewed a previous version of this manuscript and my comments as well as those of the other reviewer were addressed. I appreciate the changes and I now only have three minor remarks.

Lines 214-217:

"When both replication forks of circular chromosomes meet at ter, they collide with the divisome complex, and the XerC/XerD enzymes are activated by FtsK to resolve chromosome dimers, resulting from illegitimate recombination between left and right replicore in a fraction of the population and lethal for the cells [50]."

This is a long and complex sentence with a lot of information. I feel it might make it more clear if it is split up into two (or more) sentences.

Lines 230-232:

"Thus, the activity of these enzymes influences the composition of the DNA in OMVs, although the FtsK-XerCD complex for chromosome dimer resolution can probably not be formed."

The word "probably" in this sentence can be omitted, since the FtsK-XerCD complex can definitely not be formed when xerC or xerD has been deleted.

Line 243:

"... OMVs it not restricted to *D. shibae*, but also found in diverse genera represented by *P. marinus*, ..."

"it" should be "is".

Reviewer #2 (Comments for the Author):

Overall comments:

The author's revised article and previous rebuttal has improved the overall clarity of the main findings regarding a study of past OMV DNA sequencing studies and *E. coli* XerCD's influence on DNA release into Gram-negative bacterial OMVs from an approach using DNA sequencing of collected OMVs. Overall, the authors have addressed the majority of questions related to the aims and findings of the student in the revision. The *E. coli* xerCD study is much improved and easier to follow; however, the discussion and comparison of previously published OMV DNA sequencing data from other published species raised some additional questions that hopefully can be addressed as minor revisions. Additionally, the article still has a number of minor grammar issues throughout that would benefit from further editing (consider another grammar editing program proofread here?). For example, in nearly every paragraph, statements are missing commas, there are missing articles (e.g. the, a), run-on statements (4 lines or longer; e.g. see Lines 40-43), and other minor grammar issues.

Specific comments to address:

ABSTRACT:

Line 23-25. Typically, it's best to avoid ending on a hypothesis statement that the authors aren't pursuing. The authors should rephrase this (eg. we propose rather than we hypothesize) as a seemingly concluding and speculative statement so it avoids readers from thinking it might also be explored/ addressed in the study. A hypothesis again occurs in lines 237-238 of the manuscripts conclusions.

MAIN TEXT:

Line 44-45: "In *Pseudomonas aeruginosa*, OMVs from planktonic cultures contained plasmids [15] and chromosomal DNA [16]." Was the chromosomal DNA fully intact or just fragments?

Line 47: "Since incorporation of DNA into OMVs is so common..." this statement implies DNA is ever-present in OMVs when the evidence above suggests it can be found but not its abundance or frequency (see author's own statement made in lines 52-52 on rarity of DNA in vesicles). The authors should consider clarifying the intent here or offer evidence as to commonality/ proportion of DNA released in vesicles.

Line 81: Omit "here" from the statement, it is unnecessary and overused in the manuscript.

Lines 83-87. The *E. coli* rationale could be more concisely phrased.

Lines 91-92: Why investigate xerC and xerD? Ie. The authors should/ could state what they hoped to identify from the DNA sequencing of xer mutant vesicles.

Lines 151-152: "The vesicles were produced constitutively during exponential growth of the bacterium." It is unclear in this statement what phase of vesicles were harvested and analyzed, please clarify. If exponential phase cell-derived vesicles were collected for DNA sequencing, its is important to state when they were collected here.

Line 153-154: Why are OMV-DNA from *P. aeruginosa* biofilms being compared to presumably planktonically derived vesicles in other strains? Do OMVs from planktonic *P. aeruginosa* cells not produce OMVs with DNA? Also in methods on lines 105-106, how were OMVs purified from biofilms? Was a 1 L biofilm equivalent used here?

Lines 149-166: Can the authors briefly state how DNA results from each species' vesicles were normalized given they were from past studies? I.e. OMV particles/ml? Other?

Lines 162: "Both were completely covered with..." Unclear antecedent; what is meant by 'both' here? Chromosomes? Vibrio? Other?

Lines 165-166: Can some quantitative measurements (% value of ter/ dif from total DNA reads) from Table 1 be provided in text before this statement as to the actual proportion of ter or dif versus other DNA in the OMVs for each species OMV? What is the fraction of dif/ ter in the vesicles? Table 1 doesn't make this clear and this seems to be the implications from Fig 1.

The title of Table 1 could be improved and clarified based on footnote information provided. I.e. are the mapped reads just corresponding to ter and or dif regions? What proportion of the mapped reads were for dif or ter?

I have reviewed a previous version of this manuscript and my comments as well as those of the other reviewer were addressed. I appreciate the changes and I now only have three minor remarks.

Lines 214-217:

“When both replication forks of circular chromosomes meet at *ter*, they collide with the divisome complex, and the XerC/XerD enzymes are activated by FtsK to resolve chromosome dimers, resulting from illegitimate recombination between left and right replichore in a fraction of the population and lethal for the cells [50].”

This is a long and complex sentence with a lot of information. I feel it might make it more clear if it is split up into two (or more) sentences.

Lines 230-232:

“Thus, the activity of these enzymes influences the composition of the DNA in OMVs, although the Ftsk-XerCD complex for chromosome dimer resolution can probably not be formed.”

The word “probably” in this sentence can be omitted, since the FtsK-XerCD complex can definitely not be formed when *xerC* or *xerD* has been deleted.

Line 243:

“... OMVs **it** not restricted to *D. shibae*, but also found in diverse genera represented by *P. marinus*, ...”

“it” should be “is”.

Reviewer #1 (Comments for the Author):

I have reviewed a previous version of this manuscript and my comments as well as those of the other reviewer were addressed. I appreciate the changes and I now only have three minor remarks.

Lines 214-217:

"When both replication forks of circular chromosomes meet at ter, they collide with the divisome complex, and the XerC/XerD enzymes are activated by FtsK to resolve chromosome dimers, resulting from illegitimate recombination between left and right replicore in a fraction of the population and lethal for the cells [50]."

This is a long and complex sentence with a lot of information. I feel it might make it more clear if it is split up into two (or more) sentences.

Answer: Thank you for the suggestion. Sentences now read: "When both replication forks of circular chromosomes meet at ter, they collide with the divisome complex. Chromosome dimers, resulting from illegitimate recombination between left and right replicore in a fraction of the population, are resolved by the FtsK-activated XerC/XerD enzymes".

Lines 230-232:

"Thus, the activity of these enzymes influences the composition of the DNA in OMVs, although the Ftsk-XerCD complex for chromosome dimer resolution can probably not be formed."

The word "probably" in this sentence can be omitted, since the FtsK-XerCD complex can definitely not be formed when xerC or xerD has been deleted.

Answer: has been deleted

Line 243:

"... OMVs it not restricted to D. shibae, but also found in diverse genera represented by P. marinus, ..."

"it" should be "is".

Answer: has been corrected.

Reviewer #2 (Comments for the Author):

Overall comments:

The author's revised article and previous rebuttal has improved the overall clarity of the main findings regarding a study of past OMV DNA sequencing studies and E.coli XerCD's influence on DNA release into Gram-negative bacterial OMVs from an approach using DNA sequencing of collected OMVs. Overall, the authors have addressed the majority of questions related to the aims and findings of the student in the revision. The E. coli xerCD study is much improved and easier to follow; however, the discussion and comparison of previously published OMV DNA sequencing data from other published species raised some additional questions that hopefully can be addressed as minor revisions. Additionally, the article still has a number of minor grammar issues throughout that would benefit from further editing (consider another grammar editing program proofread here?). For example, in nearly every paragraph, statements are missing commas, there are missing articles (e.g. the, a), run-on statements (4 lines or longer; e.g. see Lines 40-43), and other minor grammar issues.

Answer: Thank you. We modified the manuscript according to your suggestions.

Specific comments to address:

ABSTRACT:

Line 23-25. Typically, it's best to avoid ending on a hypothesis statement that the authors aren't pursuing. The authors should rephrase this (eg. we propose rather than we hypothesize) as a seemingly concluding and speculative statement so it avoids readers from thinking it might also be explored/ addressed in the study. A hypothesis again occurs in lines 237-238 of the manuscripts conclusions.

Answer: to hypothesize has been replaced by to propose at both occasions.

MAIN TEXT:

Line 44-45: "In *Pseudomonas aeruginosa*, OMVs from planktonic cultures contained plasmids [15] and chromosomal DNA [16]." Was the chromosomal DNA fully intact or just fragments?

Answer: In these studies, the presence of DNA had been tested using marker genes and staining. The DNA has not been sequenced.

Line 47: "Since incorporation of DNA into OMVs is so common..." this statement implies DNA is ever-present in OMVs when the evidence above suggests it can be found but not its abundance or frequency (see author's own statement made in lines 52-52 on rarity of DNA in vesicles). The authors should consider clarifying the intent here or offer evidence as to commonality/ proportion of DNA released in vesicles.

Answer: "Since incorporation of DNA into OMVs is so common" has been deleted. The statement regarding HGT stands for itself.

Line 81: Omit "here" from the statement, it is unnecessary and overused in the manuscript.

Answer: has been removed.

Lines 83-87. The *E. coli* rationale could be more concisely phrased.

Answer: Sentence now reads: "We chose *E. coli* as an additional model because it is the archetypical, best understood organism regarding replication and cell division and a library of well-characterized gene knockouts is available, including *xerC* and *xerD*."

Lines 91-92: Why investigate *xerC* and *xerD*? I.e. The authors should/ could state what they hoped to identify from the DNA sequencing of *xer* mutant vesicles.

Answer: You are right. We added a clarifying sentence: "We hypothesized that constitutive OMV secretion in *D. shibae* is coupled to cell division and that these vesicles remove over-replicated chromosomal DNA at the end of the cell cycle, which would otherwise halt cell division and thus be lethal to the cell. The enrichment of *dif* points towards a role of XerCD in this process."

Lines 151-152: "The vesicles were produced constitutively during exponential growth of the bacterium." It is unclear in this statement what phase of vesicles were harvested and analyzed, please clarify. If exponential phase cell-derived vesicles were collected for DNA sequencing, it is important to state when they were collected here.

Answer: The authors of this study stated in their supplementary material that vesicles were harvested during mid- to late exponential phase. We added the following: "In DNA from vesicles harvested from growing cells, a broader 100 kb region around *ter* was enriched with several distinct peaks, the highest located directly at the *dif* site (Figure 1A)."

Line 153-154: Why are OMV-DNA from *P. aeruginosa* biofilms being compared to presumably

planktonically derived vesicles in other strains? Do OMVs from planktonic *P. aeruginosa* cells not produce OMVs with DNA? Also in methods on lines 105-106, how were OMVs purified from biofilms? Was a 1 L biofilm equivalent used here?

Answer: We selected all studies for which next-generation sequencing of vesicle DNA was available. Many more studies, among them also for planktonic *P. aeruginosa* vesicles detected DNA but did not perform sequencing. We made the differences to the other vesicles clear and the analysis also showed that the DNA content differs in an expected way (e.g. the typical "V"-shape from replicating chromosomal DNA). We thought it was a good idea to compare all of the few available datasets in one place.

Lines 149-166: Can the authors briefly state how DNA results from each species' vesicles were normalized given they were from past studies? I.e. OMV particles/ml? Other?

Answer: The sequencing data of each species was treated as an entity for which the relative distribution on the chromosome was analysed. Over- and underrepresentation are calculated based on the assumption that chromosomal DNA as a reference varies maximal between the *ori* and *ter* for replicating cells (which we could partially observe for the *P. aeruginosa* vesicle DNA) but otherwise evenly covered. For this purpose, no absolute quantification is necessary. Sequencing libraries are usually prepared using equimolar amounts of DNA to guarantee sufficient sequencing depths for each sample. For absolute quantification, the yield of DNA from samples processed with one defined protocol would be needed. We doubt that the comparison of samples from four different laboratories will give a meaningful result. We also think that this question is of minor importance for the analysed data.

Lines 162: "Both were completely covered with..." Unclear antecedent; what is meant by 'both' here? Chromosomes? *Vibrio*? Other?

Answer: Sentence has been rephrased: "The *V. cholerae* genome consists of two chromosomes. Both of them were completely covered by vesicle DNA, with their *dif* sites found among the highest of several distinct peaks (Figure 1C)".

Lines 165-166: Can some quantitative measurements (% value of *ter*/*dif* from total DNA reads) from Table 1 be provided in text before this statement as to the actual proportion of *ter* or *dif* versus other DNA in the OMVs for each species OMV? What is the fraction of *dif*/*ter* in the vesicles? Table 1 doesn't make this clear and this seems to be the implications from Fig 1.

Answer: We added the nine fold enrichment from the table to the text: "Both of them were completely covered by vesicle DNA, with their *dif* sites at *ter* nine fold enriched compared to the remainder of the chromosome and found among the highest of several distinct peaks (Figure 1C)."

The title of Table 1 could be improved and clarified based on footnote information provided. I.e. are the mapped reads just corresponding to *ter* and or *dif* regions? What proportion of the mapped reads were for *dif* or *ter*?

Answer: The title of table 1 has been updated: "Table 1. Strain information, mapped reads to the whole genome (total) and terminus (*ter*), summary statistics for mappings to random locations and enrichment of *ter*-located reads compared to the median along the chromosome." We added a column with enrichment of reads mapping to *ter* divided by the median of randomly mapping reads.

Re: Spectrum02343-23R3 (The Effect of site specific Recombinases XerCD on the Removal of over-replicated chromosomal DNA through Outer Membrane Vesicles in Bacteria)

Dear Dr. Jürgen Tomasch:

Your manuscript has been accepted, and I am forwarding it to the ASM production staff for publication. As you know from previous correspondence, there was a delay due to one of the reviewers's personal situation. Our sincere apologies.

Your paper will first be checked to make sure all elements meet the technical requirements. ASM staff will contact you if anything needs to be revised before copyediting and production can begin. Otherwise, you will be notified when your proofs are ready to be viewed.

Sincerely,
Silvia Cardona
Editor
Microbiology Spectrum